

# Effects of seed priming with salicylic acid on cuticular wax deposition in sweet sorghum under drought stress

Luhua Yao and Yitao Wu

Department of Agriculture and Forestry, Hainan Tropical Ocean University, Sanya, Hainan Province, China

## ABSTRACT

**Background**. Seed priming is an affordable and effective method to enhance crop drought tolerance by improving seed germination and seedling vigor. However, whether seed priming alters cuticle formation, which could contribute to drought resistance in seedlings, remains unclear.

**Materials and Methods**. In this study, sweet sorghum seeds belonging to three varieties were primed with salicylic acid (SA) and the seedlings were exposed to drought stress. The seeds were primed with SA with concentrations of 50, 100, 150, 200, 250 mg L$^{-1}$. For drought treatment (35% field capacity, 80% of field capacity as the normal condition), both SA-primed and non-primed seeds were planted in a soil mixture.

**Results**. Under drought conditions, SA priming at 150 mg L$^{-1}$ significantly increased total wax content (12.3%–33.3%), primary alcohol content (42.0%–57.0%), proline content (70.9%–119%), leaf water content (9.8%–36.6%), water use efficiency (28.4%–120%), and biomass (25%–30%). Additionally, leaf water loss rate and chlorophyll leaching rate were significantly reduced. Correlation analysis revealed strong positive associations ($r = 0.79$–$0.86$, $P < 0.001$) between total wax content and water status as well as proline content. Furthermore, under both well-watered and drought conditions, SA priming at 150 mg L$^{-1}$ significantly increased leaf wax content and biomass in all sorghum varieties.

**Conclusion**. Seed priming with salicylic acid at 150 mg L$^{-1}$ not only promotes normal growth under well-watered conditions, but also enhances sorghum's adaptability to drought stress, ultimately contributing to drought tolerance without penalizing growth.

Corresponding author
Luhua Yao, 1979916660@qq.com

## INTRODUCTION

Drought, a major abiotic stress intensified by global climate change, severely impacts agriculture (*Hadebe, Modi & Mabhaudhi, 2017*). Global water shortages due to drought have reduced crop yields on 49% of farmland (*Rosa et al., 2020*; *Vogel et al., 2019*). Numerous studies have highlighted that seedling growth is crucial for crop yield under stress conditions (*Hubbard, Germida & Vujanovic, 2012*). Therefore, improving crop resistance at the seedling stage is essential for stable yields. Seed priming has reportedly emerged as an effective, low-cost method to boost seedling growth and resistance (*Johnson & Puthurb, 2021*; *Rhaman et al., 2021*). Specifically, hormone activation rapidly stimulates metabolic

and stress-response processes in plants (*Rhaman et al., 2021*). Salicylic acid (SA), a phenolic plant hormone, regulates plant growth, development, and physiological processes like photosynthesis, respiration, transpiration, and ion transport (*Khan, Prithiviraj & Smith, 2003*). Seed priming with salicylic acid (SA) has been shown to enhance drought tolerance by activating early stress-response mechanisms, such as antioxidant defense systems, osmotic adjustment, and stomatal regulation (*Rhaman et al., 2021*). For example, in rice, SA priming enhanced seedling growth and water-use efficiency under water-deficient conditions (*Nie et al., 2022*). As a seed initiator, SA can modulate lipid peroxidation and proline content to enhance abiotic stress resistance (*Ren et al., 2018*). However, the effectiveness of SA in improving stress resistance and yield depends on its concentration. For example, SA priming at 0.25 mM and 0.50 mM enhances rice's resistance to chromium stress by regulating ion homeostasis and nutrient absorption (*Shinwari et al., 2015*). Priming with 0.5 mM SA reportedly improved corn seed germination and seedling growth (*Li et al., 2017*). Additionally, 100 µM SA priming promoted photosynthesis, chlorophyll biosynthesis, and nutrient absorption in alfalfa under salt and iron deficiency stresses (*Boukari et al., 2019*). Therefore, selecting the optimal priming concentration is crucial for effectively enhancing plant resistance. In addition to drought, SA seed priming has been shown to mitigate various abiotic stresses, including salinity, heavy metal toxicity, and extreme temperatures. For instance, salicylic acid seed priming modulates total chlorophyll, soluble sugars, proteins to improve germination and seedling growth of salt stressed wheat (*Azeem et al., 2019*). These findings underscore the versatility of SA seed priming as a tool for improving plant resilience under multiple abiotic stress conditions.

Cuticular wax, a protective barrier on plant surfaces against external stress, primarily consists of very long-chain fatty acids, including alkanes, alcohols, aldehydes, and other compounds (*Yeats & Rose, 2013*). Changes in cuticular wax can effectively reduce plant water loss (*Zhang et al., 2019*). For instance, an increase in cuticular wax content, particularly non-polar alkanes, significantly decreases the rate of water loss in plants (*Bourdenx et al., 2011*). Additionally, wax deposition influences the permeability of plant cell membranes. For example, as alkanes and aldehyde levels rise, chlorophyll leaching rate significantly decreases (*Tomasi et al., 2017*). Furthermore, alterations in cuticular wax composition affect photosynthesis. Increased wax content enhances cellular gas exchange, improving both photosynthetic and water utilization efficiency (*Su et al., 2020*). However, whether seed priming affects cuticle deposition during seedling growth and contributes to drought tolerance remains unclear.

Sorghum (*Sorghum bicolor* L.) is a versatile cereal crop with notable drought tolerance (*Hadebe, Modi & Mabhaudhi, 2017*). Our previous study found that cuticular wax deposition in sorghum seedlings enhances drought resistance, though this trait varies across different varieties (*Zhang et al., 2021*). Additionally, exogenous SA application reportedly alters the cuticular wax composition in rapeseed, thereby affecting its permeability (*Yuan et al., 2020*). Therefore, we hypothesized that SA pretreatment of sorghum seeds could modify wax deposition in the seedling cuticle, thereby enhancing its drought resistance and biomass accumulation. The novelty of this study lies in the use of SA seed priming to enhance drought resilience through the modulation of cuticular wax deposition.

While previous studies have explored SA priming in other crops, our work is the first to systematically investigate the effects of SA priming on cuticular wax biosynthesis and its role in improving drought tolerance in sorghum. This approach provides new insights into the mechanisms of SA-mediated drought tolerance and highlights the potential of cuticular wax as a key trait for improving water-use efficiency in sorghum under water-limited conditions. The present study aims to uncover how SA seed priming enhances growth and drought resistance in sweet sorghum seedlings. It thoroughly examines physiological and morphological parameters, scrutinizing their influence on plant growth and development under normal growth condition and drought stress.

## MATERIALS & METHODS

### Plant materials, plant growth conditions, SA priming and stress treatments

To ensure that our findings are directly applicable to real-world agricultural practices and to enhance the practical relevance and scalability of the results, we selected two commercial varieties (Hunnigreen and Mule 8000) widely cultivated in drought-prone regions. Seeds of the commercial varieties Hunnigreen (DLS) and Mule 8000 (ML8000) were provided by Beijing Sanye Grass Seed Lawn Co., Ltd., while the inbred sweet sorghum (P05206) seeds came from the Institute of Sorghum Research, Shanxi Academy of Agricultural Science, China. The seeds were sterilized in 3% $H_2O_2$ for 15 min, rinsed with distilled water to remove disinfectant residue, and then the seeds were primed with salicylic acid (SA) with concentrations of 50, 100, 150, 200, or 250 mg $L^{-1}$ at 20 °C in darkness for 24 h, and dried back to their original dry weight at room temperature (*Sheteiwy et al., 2016*; *Khan et al., 2015*; *Shakirova et al., 2003*). The seeds without priming were considered as the control. For drought treatment, both SA-primed and non-primed seeds were planted in a soil mixture composed of peat and arable soil in a 1:2 ratio. The soil had a pH of 6.57 and contained 85 mg $kg^{-1}$ of alkaline dispelled nitrogen, 24 mg $kg^{-1}$ of available phosphorus, and 114 mg $kg^{-1}$ of available potassim. Each pot (15 cm × 20 cm) was filled with 2.5 kg steam-sterilized soil. These pots were then positioned in a greenhouse with daytime temperatures maintained at 28 °C and nighttime temperatures at 20 °C. They were watered every three days to maintain soil moisture at approximately 80% of field capacity. When the seedlings reached the three-leaf stage, plants from non-primed seeds and SA-primed seeds were exposed to 14 days of drought stress (maintained at 35% of field capacity). The soil moisture reduction to 35% field capacity over 2 days. Four replicates (pots) were established for each treatment, with three seedlings per pot.

When the seedlings reached the three-leaf stage, plants from non-primed seeds were exposed to 14 days of drought stress (maintained at 35% of field capacity), designated as the D treatment. Similarly, plants from SA-primed seeds underwent the same 14-day drought stress, representing the combined SA&D treatment. Plants originating from SA-primed seeds and grown under well-watered conditions were designated as the SA treatment, while those from non-primed seeds, also under well-watered conditions (at 80% field capacity), served as the control (CK). Water was replenished daily to maintain the required soil

moisture levels. Fourteen days after drought treatment, the fully expanded leaf from the top was sampled for cuticular wax, physiological, and biomass analyses.

## Leaf cuticle composition and structure analysis
### Leaf wax extraction
*Wax extraction.* Leaves from each replicate were immersed in 10 mL chloroform with 1 μg tetracosane as an internal standard for 30 s, and the process was repeated twice. Prior to extraction, leaf surface areas were measured using the WinFOLIA professional leaf image analysis system (Regent Instrument Inc., Canada) and a digitizing scanner (EPSON V750, Japan) (*Cawkell, 1991*).

The wax extracts were dried under nitrogen at 40 °C, then derivatized with 20 μL pyridine and 20 μL BSTFA for 45 min at 70 °C (*Buschhaus & Jetter, 2012*; *Yao et al., 2019*). Excess BSTFA was evaporated under nitrogen, and the sample was re-dissolved in one mL hexane for 9790 II gas chromatography (Fu-Li, China) and GCMS-QP2010 Ultra Mass Spectrometric Detector (Shimadzu Corp., Kyoto, Japan) analysis.

### GC and GC/MS analysis
The gas chromatography (GC) analysis was performed using a DM-5 capillary column (30 m × 0.32 mm × 0.25 μm; Dikma Technologies Inc., USA) with nitrogen as the carrier gas. The injection port and flame ionization detector (FID) were maintained at 300 °C and 320 °C, respectively. The oven temperature program was initiated at 80 °C, then ramped up to 260 °C at a rate of 15 °C min$^{-1}$, held for 10 min, followed by a gradual increase to 290 °C at 2 °C min$^{-1}$, and finally raised to 320 °C at 5 °C min$^{-1}$ with a 10-min hold time.

For compound identification, the samples were subsequently analyzed by gas chromatography-mass spectrometry (GC-MS) using a QP2010 Ultra system equipped with an HP-5 MS capillary column (30 m × 0.32 mm × 0.25 μm) and helium as the carrier gas. The identification of compounds was achieved by matching their mass spectra with reference data and authentic standards. The quantification of leaf cuticular waxes was determined using an internal standard method, and the results were expressed in micrograms per square centimeter (μg cm$^{-2}$).

## Leaf growth and physiological characteristic analysis
### Photosynthesis, water use efficency and chlorophyll content analysis
Photosynthetic rates (Pn), stomatal conductance (Gs), intercellular CO2 concentrations (Ci), and transpiration rates (Tr) were measured on the second expanded leaf from the plant top between 9:00 and 11:00 a.m. using a portable photosynthesis system (LCpro-SD, ADC Bioscientific, London, UK) with a photosynthetic photon flux of 1,000 μmol m$^{-2}$ s$^{-1}$. Leaf water use efficiency (WUE) is calculated as the ratio of Pn (net photosynthetic rate) to Tr (transpiration rate) (*Jones, 1992*). Chlorophyll content was determined using Soil Plant Analysis Development (SPAD) on the same leaf for each replicate.

### Leaf relative water content
Relative water content (RWC) was assessed according to *Barrs & Weatherley (1962)*. The first fully expanded leaves from the top were immediately weighed to obtain fresh weight (FW). After immersion in distilled water for 3 h, saturated weight (SW) was recorded,

followed by drying at 70 °C for 24 h to determine dry weight (DW). RWC was calculated as: RWC = (FW-DW)/(SW-DW) × 100%.

### Leaf water loss rate under dark condition

Leaf water loss rate under dark condition was assessed according to *Burkhardt & Pariyar (2014)*. The first fully expanded leaves from the top were immediately weighed to obtain fresh weight (FW). The plants were acclimated to darkness for 12 h to ensure stomatal closure leaf samples were then immersed in distilled water for 1 h and weighed (SW). Subsequently, the samples were placed in a dark chamber for continuous dehydration, with their weights measured at 15-minute intervals over 150 min to monitor water loss. The leaves were then dried at 70 °C for 24 h and weighed (DW). Water loss percentage was calculated using the formula: Water loss (%) = (SW-FW)/(SW-DW) × 100%.

### Chlorophyll leaching rate

Chlorophyll leaching rate (CLR) were performed as described previously (*Liu et al., 2023*). Healthy, fully-expanded leaves (second from the apex) of sweet sorghum were collected and cut into three cm segments. The leaf segments were immersed in 30 mL of 80% ethanol and gently agitated in complete darkness. Aliquots (three mL) of the leaching solution were sampled at 20, 60, 120, 180, and 240 min intervals for spectrophotometric analysis. Absorbance measurements were performed at $\lambda$664 nm and $\lambda$647 nm using a spectrophotometer. Three biological replicates were performed for each treatment condition.

Chlorophyll leaching content (CLC, mmol $g^{-1}$) = 7.93A664+19.53A647
Chlorophyll leaching rate (%) = (CLCt/CLC$_{240}$) × 100%.
$t = 20, 60, 120, 180$ min.

### Proline and malondialdehyde contents

Leaf proline content was determined by sulfosalicylic acid spectrophotometry (*Bates, Waldren & Teare, 1973*). Briefly, 0.1 g of fresh leaf tissue was homogenized in five mL of 3% sulfosalicylic acid, followed by extraction in a boiling water bath for 10 min. After cooling, four mL toluene was added to the mixture, which was vigorously shaken for 30 s and then allowed to separate for 2 h. The absorbance of the toluene phase containing proline was measured at 520 nm using a spectrophotometer. Lipid peroxidation was quantified as malondialdehyde (MDA) content, measured using the thiobarbituric acid method (*Wang et al., 2010*). Fresh root and leaf samples (0.1 g) were homogenized in one mL of 5% (w/v) trichloroacetic acid (TCA) and centrifuged at 1,200 rpm for 15 min. Subsequently, 0.3 mL of the supernatant was mixed with 0.5 mL of 0.5% TBA, heated at 98 °C for 20 min, and then cooled. After centrifugation, absorbance was measured at 532, 600, and 450 nm using a spectrophotometer. MDA concentration was calculated and expressed as nmol $g^{-1}$ fresh weight.

## Biomass analysis

At the end of drought treatments, the shoots were harvested and dried at 70 °C for 72 h to determine total dry weight.

## Statistical analysis

All data are presented as means $\pm$ standard error (SE) ($n = 4$). To evaluate the effects of seed priming on sorghum growth and drought tolerance, we used one-way analysis of variance (ANOVA) to compare the means of different groups under normal or drought conditions. This approach was chosen to focus on the impact of seed priming, rather than the interaction effects of water status and priming treatments. *Post-hoc* analysis was performed using the least significant difference (LSD) test to identify specific differences between groups. All statistical analyses were conducted using (SPSS 17.0, Chicago), and a significance level of $p < 0.05$ was adopted. Spearman correlation analysis was performed to assess relationships between variables. To account for multiple comparisons, the Bonferroni correction was applied, adjusting the significance threshold from 0.05 to 0.01 (0.05 divided by five tests). Only correlations with $p < 0.01$ were considered statistically significant.

## RESULTS

### Seed priming with different SA concentrations improved leaf wax content of sorghum seedlings

In a preliminary experiment, sweet sorghum seeds primed with different SA concentrations were subjected to drought stress at the seedling stage (Fig. 1). Interestingly, under both normal and drought conditions, as the SA priming concentration increased, the total wax content of leaves from different varieties initially increased and then decreased, with the highest wax content observed at the 150 mg L$^{-1}$ treatment. In the varieties, Hunnigreen, Mule 8000, and P05206, the range of wax content changes due to SA priming under normal conditions was 1.67 to 1.92, 1.77 to 2.27, and 1.37 to 2.35 μg cm$^{-2}$, respectively. The total wax content of these three sorghum varieties peaked at 150 mg L$^{-1}$, representing increases of 13.6%, 15.1%, and 13.8% compared to their CK, respectively. Under drought conditions, the total wax content ranged from 1.75 to 2.00, 2.01 to 3.13, and 1.86 to 2.75 μg cm$^{-2}$, respectively, with the highest values also observed at 150 mg L$^{-1}$. In these conditions, the increases in wax content compared to CK were 13.2%, 33.3%, and 12.3%, respectively. Overall, priming seeds with 150 mg L$^{-1}$ SA significantly increased the total wax content in sorghum leaves under both normal and drought conditions.

### Seed priming with different SA concentrations improved height and boimass of sorghum seedlings

To explore how drought resistance was affected by SA priming at different concentrations, phenotypic and physiological indices were measured (Figs. 2–5). Seed priming improved the height and biomass of sorghum seedlings (Fig. 2). Under both normal and drought conditions, increasing SA concentrations initially led to increases in height and biomass, followed by a decline at higher concentrations (Fig. 2). The height and biomass of all three varieties increased significantly at lower concentrations, particularly at 100 and 150 mg L$^{-1}$ (Figs. 2A–2C).

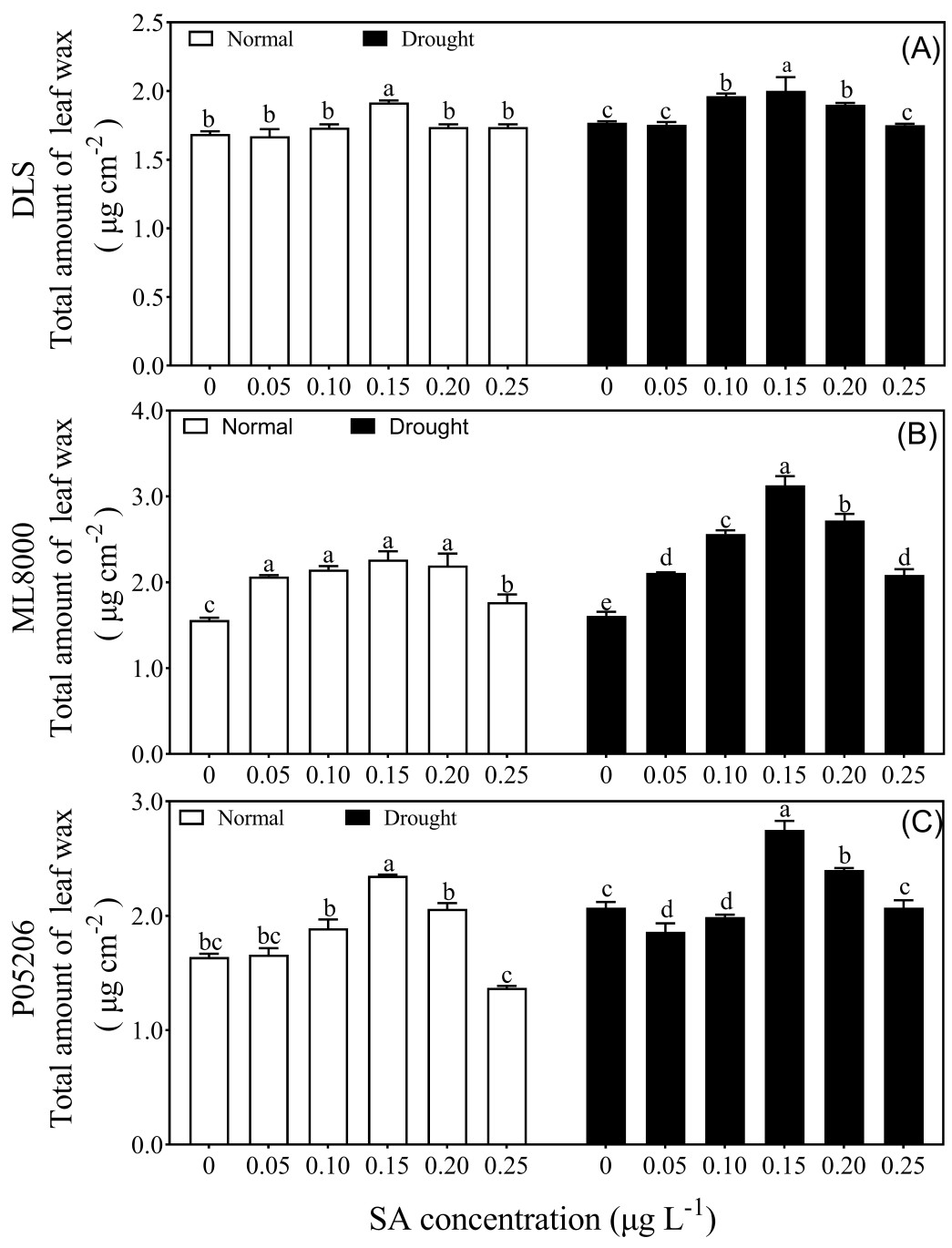

**Figure 1  Effects of SA priming with different concentrations on leaf wax of DLS (A), ML8000 (B) and P05206 (C).** DLS, Hunnigreen; ML8000, Mule 8000; P05206, the inbred sweet sorghum. Normal, plants under well-watered condition; drought, plants subjected to drought stress. Different lowercase letters within same parameters among different treatments represented significance according to the least significant difference test ($P < 0.05$).

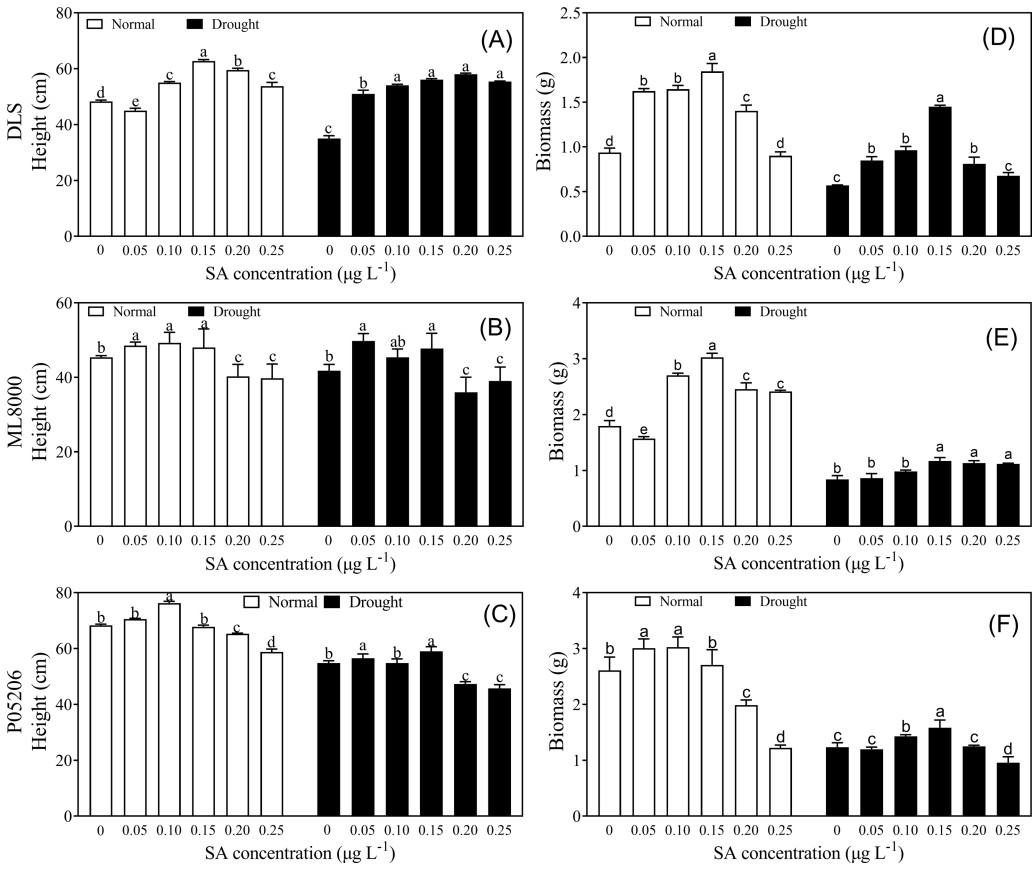

**Figure 2  Effects of SA priming with different concentrations on height and biomass of DLS, ML8000 and P05206.** (A) Height of DLS seedlings. (B) Height of ML8000 seedlings. (C) Height of P05206 seedlings. (D) Biomass of DLS seedlings. (E) Biomass of ML8000 seedlings. (F) Biomass of P05206 seedlings. DLS, Hunnigreen; ML8000, Mule 8000; P05206, the inbred sweet sorghum. Normal, plants under well-watered condition; drought, plants subjected to drought stress. Different lowercase letters within same parameters among different treatments represented significance according to the least significant difference test ($P < 0.05$).

## Seed priming with different SA concentrations influenced photosynthetic indices of sorghum seedlings

Seed priming can improve the photosynthetic indices of sorghum seedlings under both normal and drought conditions (Fig. 3). With increasing concentrations of SA priming, the net photosynthetic rate, stomatal conductance, and intercellular carbon dioxide levels in the leaves of different varieties followed a trend of first increasing and then decreasing, except for the transpiration rate. The net photosynthetic rate significantly increased in drought-stressed seedlings pretreated with 150 mg L$^{-1}$ SA compared to untreated drought-stressed seedlings.

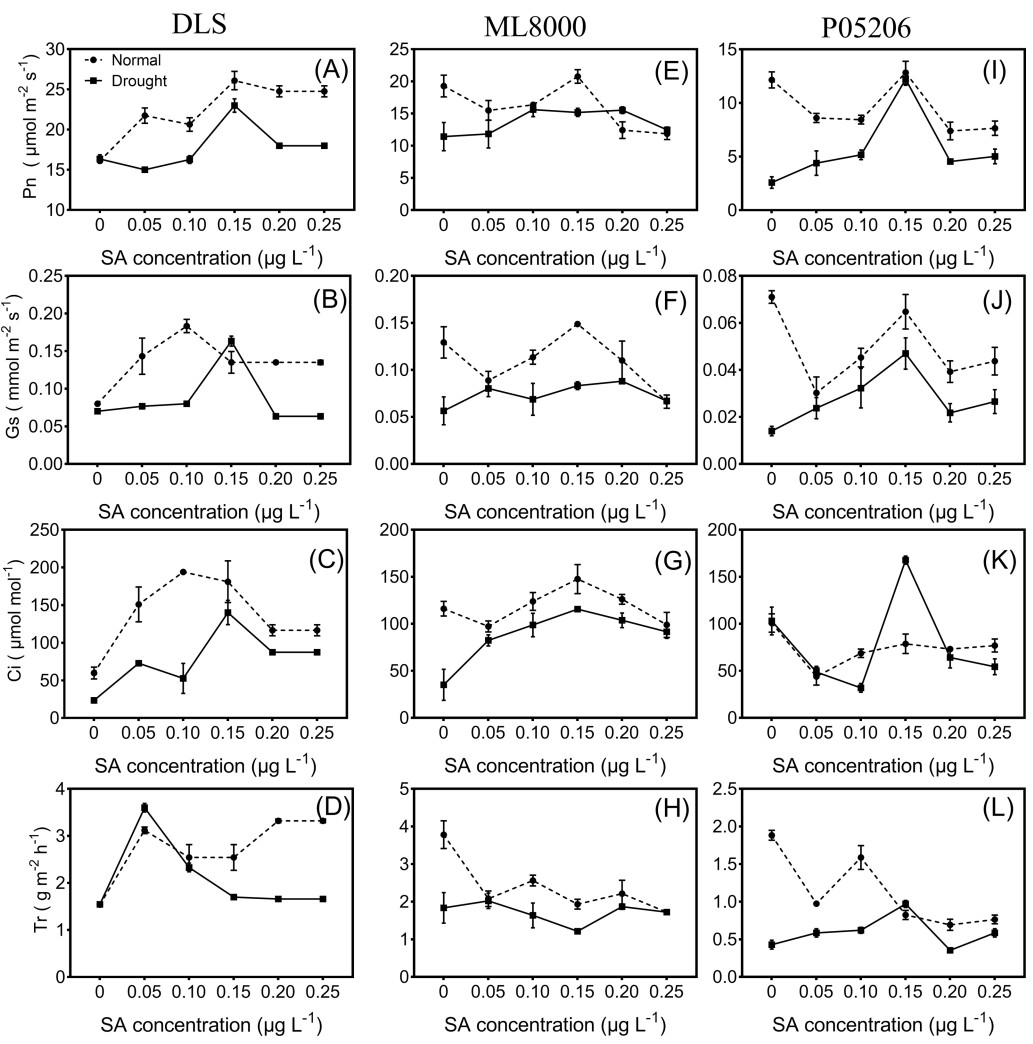

**Figure 3** **Effects of SA priming with different concentrations on photosynthetic indexes of DLS, ML8000 and P05206.** (A) (B) (C) and (D) Photosynthesis parameters of DLS. (E) (F) (G) and (H) Photosynthesis parameters of ML8000. (I) (J) (K) and (L) Photosynthesis parameters of P05206. DLS, Hunnigreen; ML8000, Mule 8000; P05206, the inbred sweet sorghum. Pn, net photosynthetic rate; Gs, stomatal conductance; Ci, intercellular $CO_2$ concentration; Tr, transpiration rate. Normal, plants under well-watered condition; Drought, plants subjected to drought stress. Different lowercase letters within same parameters among different treatments represented significance according to the least significant difference test ($P < 0.05$).

## Seed priming with different SA concentrations influenced water status of sorghum seedlings

Notably, the relative water content and water use efficiency of the different varieties also initially increased and then decreased with increasing SA priming concentrations under both normal and drought conditions (Fig. 4). The highest values were observed at 150 mg L$^{-1}$ (Fig. 4). Specifically, under drought stress, the 150 mg L$^{-1}$ SA priming treatment resulted in the highest water content in the leaves of all varieties, with Hunnigreen's water content increasing by 9.8% compared to the control, Mule 8000 by 36.6%, and P05206

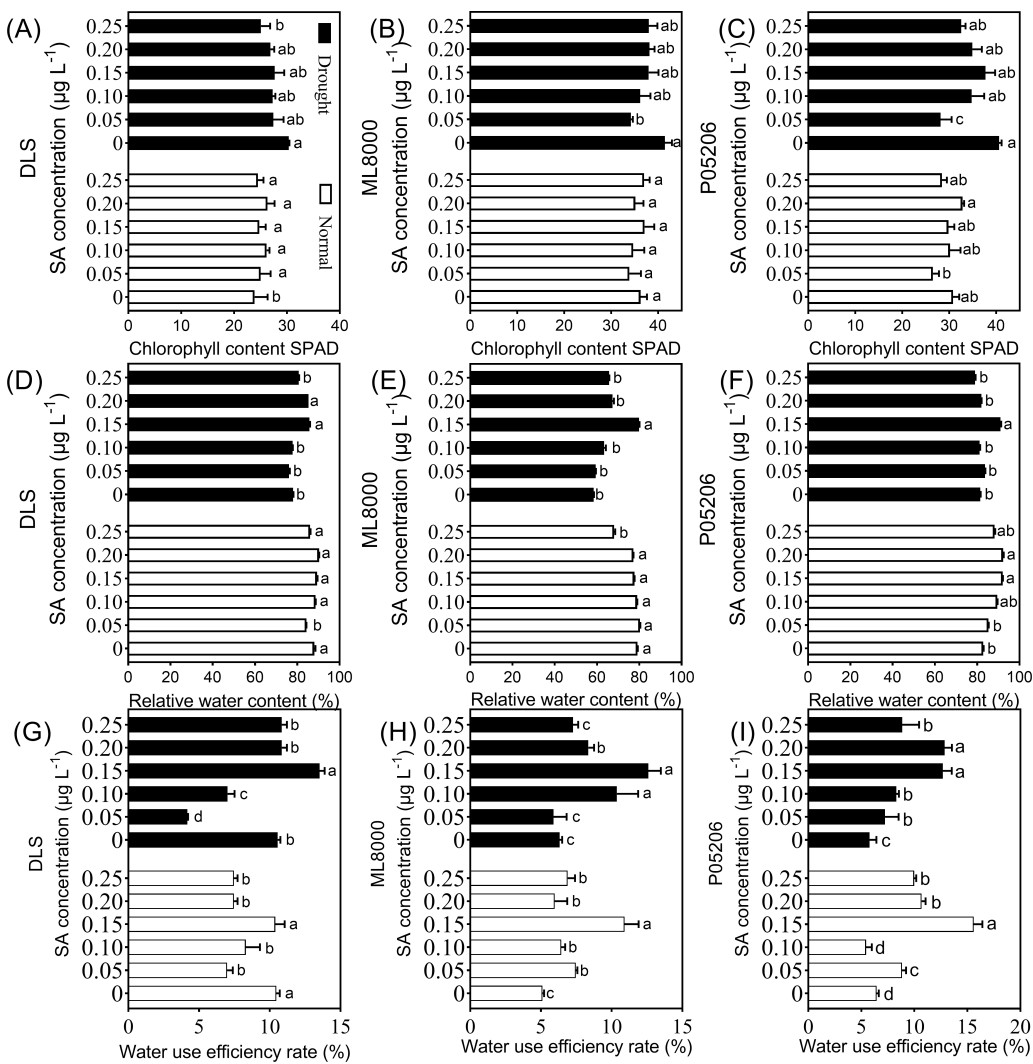

**Figure 4** Effects of SA priming with different concentrations on chlorophyll content (SPAD value), relative water content and water use efficiency. (A) (B) and (C) The chlorophyll content of DLS, ML8000 and P05206 seedlings treated with different SA priming concentration treatment. (D) (E) and (F) The relative water content of DLS, ML8000 and P05206 seedlings treated with different SA priming concentration treatment. (G) (H) and (I) The water use efficiency rate of DLS, ML8000 and P05206 seedlings treated with different SA priming concentration treatment. DLS, Hunnigreen; ML8000, Mule 8000; P05206, the inbred sweet sorghum. Normal, plants under well-watered condition; Drought, plants subjected to drought stress. Different lowercase letters within same parameters among different treatments represented significance according to the least significant difference test ($P < 0.05$).

by 11.6%. Additionally, the water use efficiency of Hunnigreen was significantly enhanced compared to the control seedlings. The highest water use efficiency was observed in the 150 mg L$^{-1}$ SA treatment, with an increase of 28.4% in Hunnigreen, 99.1% in Mule 8000, and a doubling in P05206 compared to the controls. In summary, SA priming affects leaf relative water content and water use efficiency, with the 150 mg L$^{-1}$ treatment showing consistent and optimal effects across various indicators and varieties.

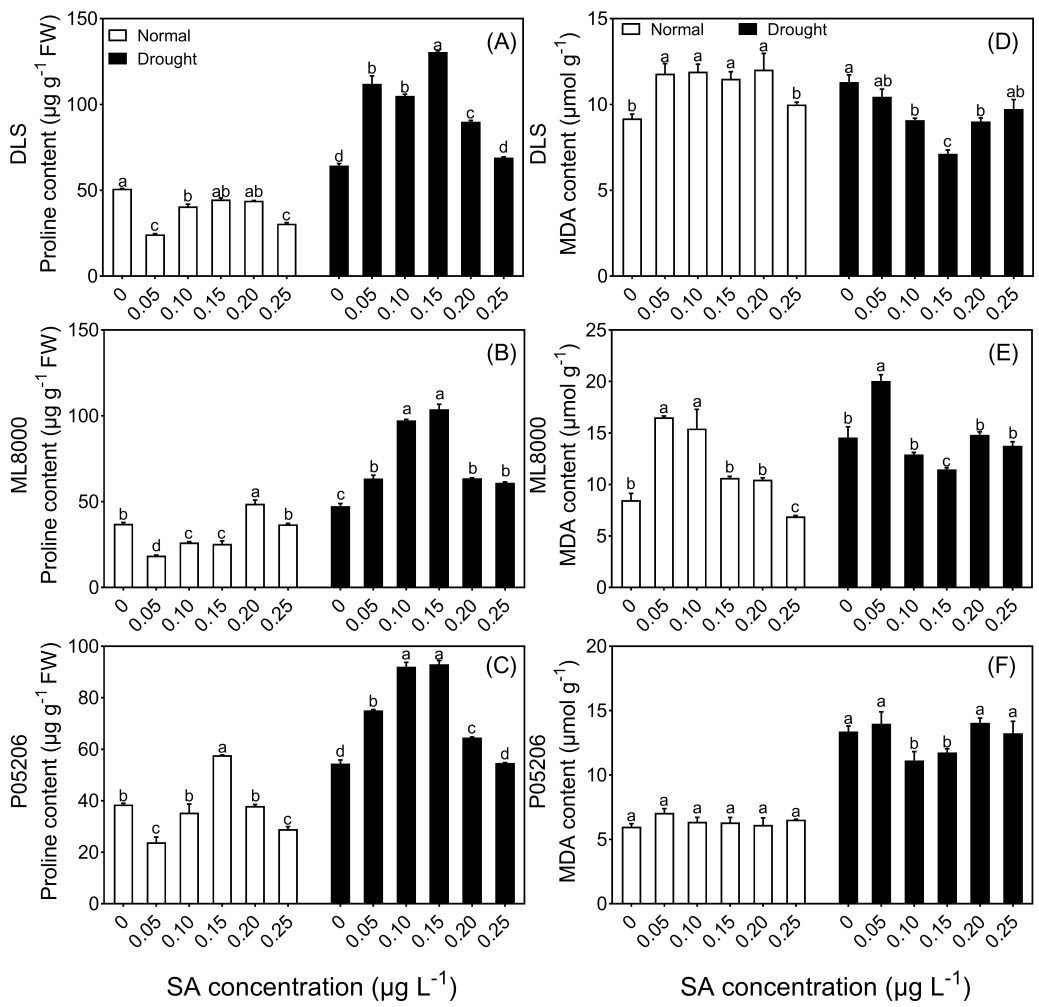

**Figure 5 Effects of SA priming with different concentrations on malondialdehyde and proline content.** (A) (B) and (C) The proline content of DLS, ML8000 and P05206 seedlings treated with different SA priming concentration treatment. (D) (E) and (F) MDA content of DLS, ML8000 and P05206 seedlings treated with different SA priming concentration treatment. DLS, Hunnigreen; ML8000, Mule 8000; P05206, the inbred sweet sorghum. Normal, plants under well-watered condition; Drought, plants subjected to drought stress. Different lowercase letters within same parameters among different treatments represented significance according to the least significant difference test ($P < 0.05$).

## Seed priming with different SA concentrations influenced the content of proline and MDA of sorghum seedlings

Under both normal and drought conditions, the proline content in the leaves of three sorghum varieties initially increased and then decreased with increasing SA-priming concentrations (Figs. 5A, 5B, 5C). Under normal conditions, the proline content in Mule 8000, primed with 200 mg $L^{-1}$ SA, increased by 31.5% compared to the non-primed seedlings. In P05206, a 150 mg $L^{-1}$ SA priming treatment resulted in a 49.5% increase compared to the non-primed seedlings. Under drought conditions, both 100 mg $L^{-1}$ and 150 mg $L^{-1}$ SA priming significantly increased proline content across all three varieties,

with increases of 63% and 103%, respectively, compared to the non-primed Hunnigreen seedlings (Fig. 5A), 105% and 119% in Mule 8000 (Fig. 5B), and 69.0% and 70.9% in P05206 (Fig. 5C).

Additionally, under drought conditions, malondialdehyde content initially decreased and then increased with increasing SA priming concentrations (Figs. 5D, 5E, 5F). Notably, in the 150 mg $L^{-1}$ SA-primed treatment, malondialdehyde content was significantly lower than the control, with decreases of 36.9% in Hunnigreen and 13.8% in Mule 8000 compared to the non-primed seedlings.

Through the analysis of sorghum wax, growth, and physiological indicators induced by different concentrations of SA priming, it was found that 150 mg $L^{-1}$ SA had a positive effect on the cuticle wax, growth, and resistance of all three varieties, with the greatest impact. Therefore, this concentration was chosen for the analysis of wax content, its components, and the drought resistance of the cuticle.

### Effect of SA priming on cuticular wax components of sorghum

To explore how leaf cuticle components were affected by SA priming under drought conditions, the amount and composition of leaf cuticular wax and cutin were further examined (Fig. 6). The total wax content was significantly increased in seedlings pretreated with SA compared to control seedlings under both normal and drought conditions (Figs. 6A, 6B, 6C). The cuticular waxes in sorghum leaves consisted of alkanes, alcohols, aldehydes, and triterpenoids. Under normal conditions, SA priming increased the total wax content of Hunnigreen by 13.7%, Mule 8000 by 15.1%, and P05206 by 13.8%. The SA priming also significantly affected wax composition. In SA-primed seedlings, alkane content in Mule 8000 increased by 63.1%, and primary alcohol content in Hunnigreen nearly doubled. The $\alpha$-amyrin content in both Hunnigreen and Mule 8000 increased by 1.1 and 1.2 times, respectively. Salicylic acid priming had no significant effect on other wax components in the sorghum varieties. While total wax content under drought stress increased, there was no significant difference compared to the CK. However, in seedlings treated with SA priming under drought stress, the total wax, primary alcohol, aldehyde, and $\alpha$-amyrin content were significantly increased compared to drought-stressed seedlings without SA priming. In Hunnigreen, the content increased by 13.2%, 57.0%, 38.9%, and 77.1%, respectively (Fig. 6A). In Mule 8000, the increases were 33.3%, 42.0%, 164.7%, and 257.9%, respectively (Fig. 6B). In P05206, the wax and primary alcohol content increased significantly by 12.3% and 50.2%, respectively (Fig. 6C).

### Effect of SA priming on the cuticular wax carbon chain distribution of sorghum

The distribution range of alkane carbon chains in different sorghum varieties is $C_{25}$–$C_{37}$ (Fig. 7). The results indicated that the content of $C_{27}$ and $C_{29}$ alkanes was significantly influenced by SA priming, though no consistent changes were observed across different varieties (Figs. 7A, 7B, 7C). The distribution range of primary alcohol carbon chains in different sorghum varieties is $C_{26}$–$C_{30}$ (Figs. 8A, 8B, 8C). The content of primary alcohols in the $C_{26}$ and $C_{28}$ carbon chains of Hunnigreen and Mule 8000 was significantly

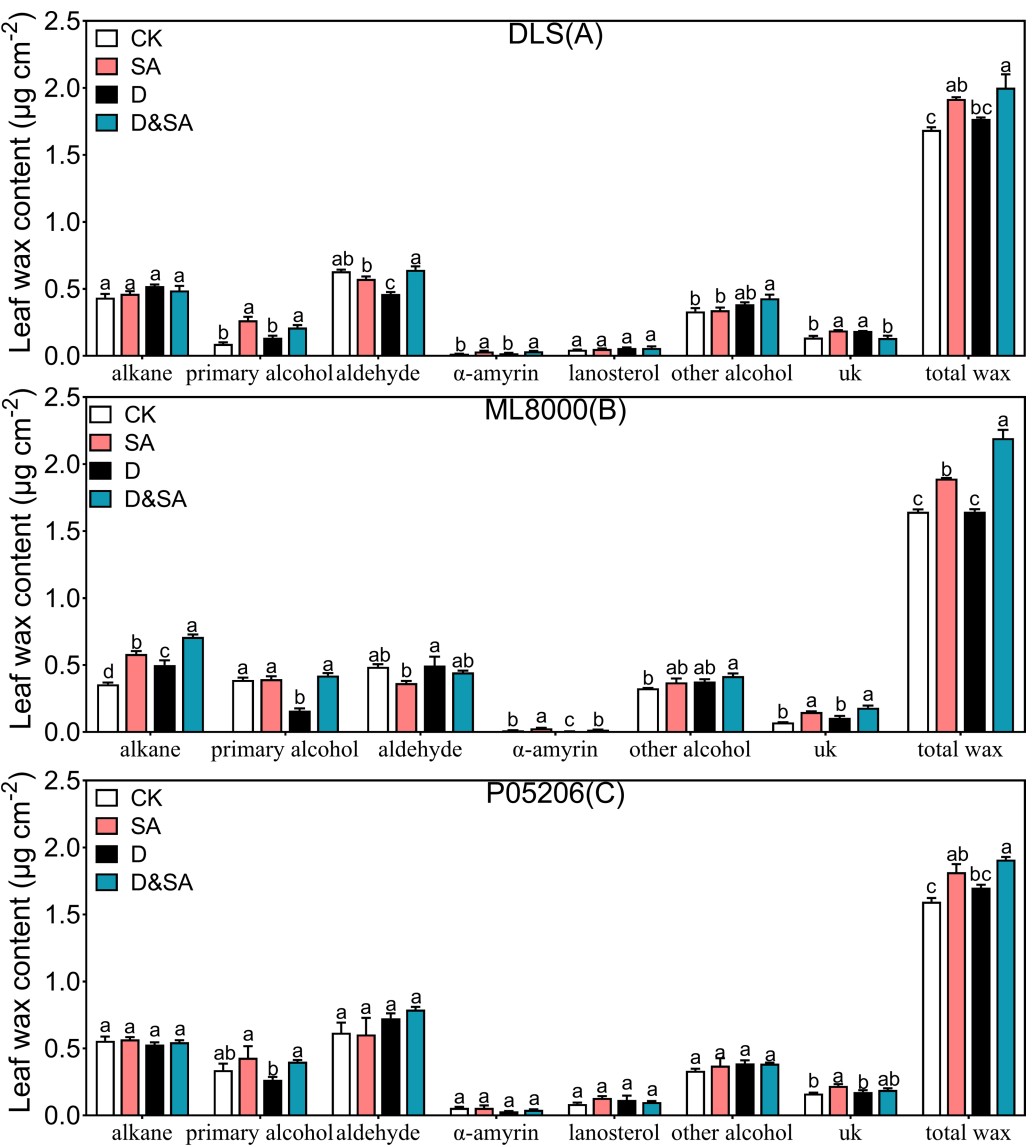

**Figure 6** **Effects of seed priming with SA on total cuticular wax and wax composition.** (A), total cuticular wax and wax composition of DLS; (B), total cuticular wax and wax composition of ML8000; (C), total cuticular wax and wax composition of P05206. CK, plants from non-primed seeds under well-watered condition; D, plants from non-primed seeds subjected to drought stress; SA, plants from SA-primed seeds under well-watered condition, and SA&D, plants from SA-primed seeds subjected to drought stress. Different lowercase letters within same parameters among different treatments represented significance according to the least significant difference test ($P < 0.05$). The data were means $\pm$ SE ($n = 4$). uk, unkown component.

increased under drought conditions (Figs. 8A, 8B). The distribution range of aldehyde carbon chains in different sorghum varieties is $C_{26}$–$C_{34}$. Under drought conditions, SA priming significantly reduced the $C_{30}$ aldehyde content in Hunnigreen (Figs. 8D, 8E, 8F). In Mule 8000, primed seedlings under drought stress exhibited a significant decrease in C28 aldehyde content and a significant increase in $C_{26}$ aldehyde content (Fig. 8E).

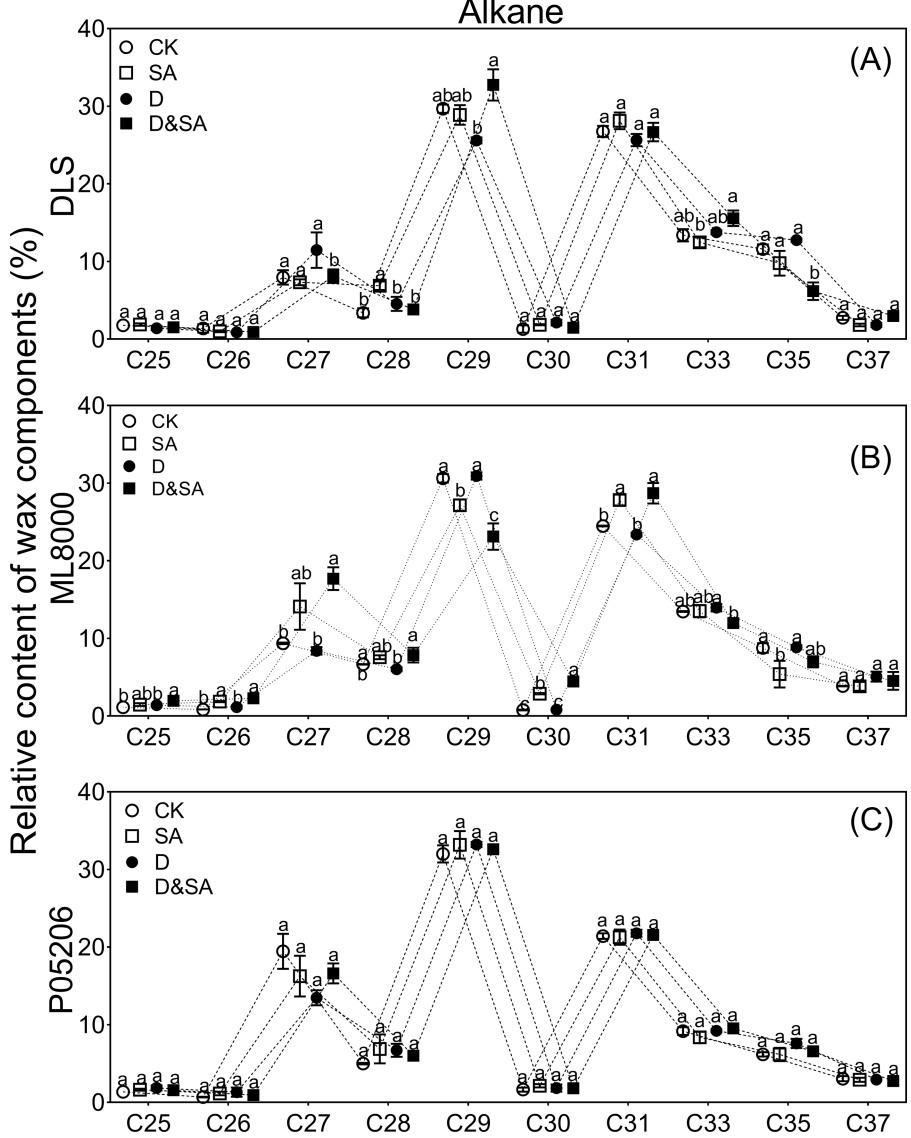

**Figure 7 Effects of SA priming on the carbon chain distribution of alkane.** (A) (B) and (C) The chain length of n-alkanes. The data were means ± SE ($n = 3$). CK, plants from non-primed seeds under well-watered condition; D, plants from non-primed seeds subjected to drought stress; SA, plants from SA-primed seeds under well-watered condition, and SA&D, plants from SA-primed seeds subjected to drought stress. Different lowercase letters within same parameters among different treatments represented significance according to the least significant difference test ($P < 0.05$). DLS, Hunnigreen; ML8000, Mule 8000; P05206, the inbred sweet sorghum.

## Effects of SA priming on water loss rate and chlorophyll leaching rate of Sorghum

To gain insight into the drought resistance of the cuticle in response to SA priming, we analyzed the water loss rate and chlorophyll leaching rate (Fig. 9). Salicylic acid priming reduced the leaf water loss rate in Hunnigreen under drought conditions (Fig. 9A) and in Mule 8000 and Pin05206 under both normal and drought conditions (Figs. 9B, 9C).

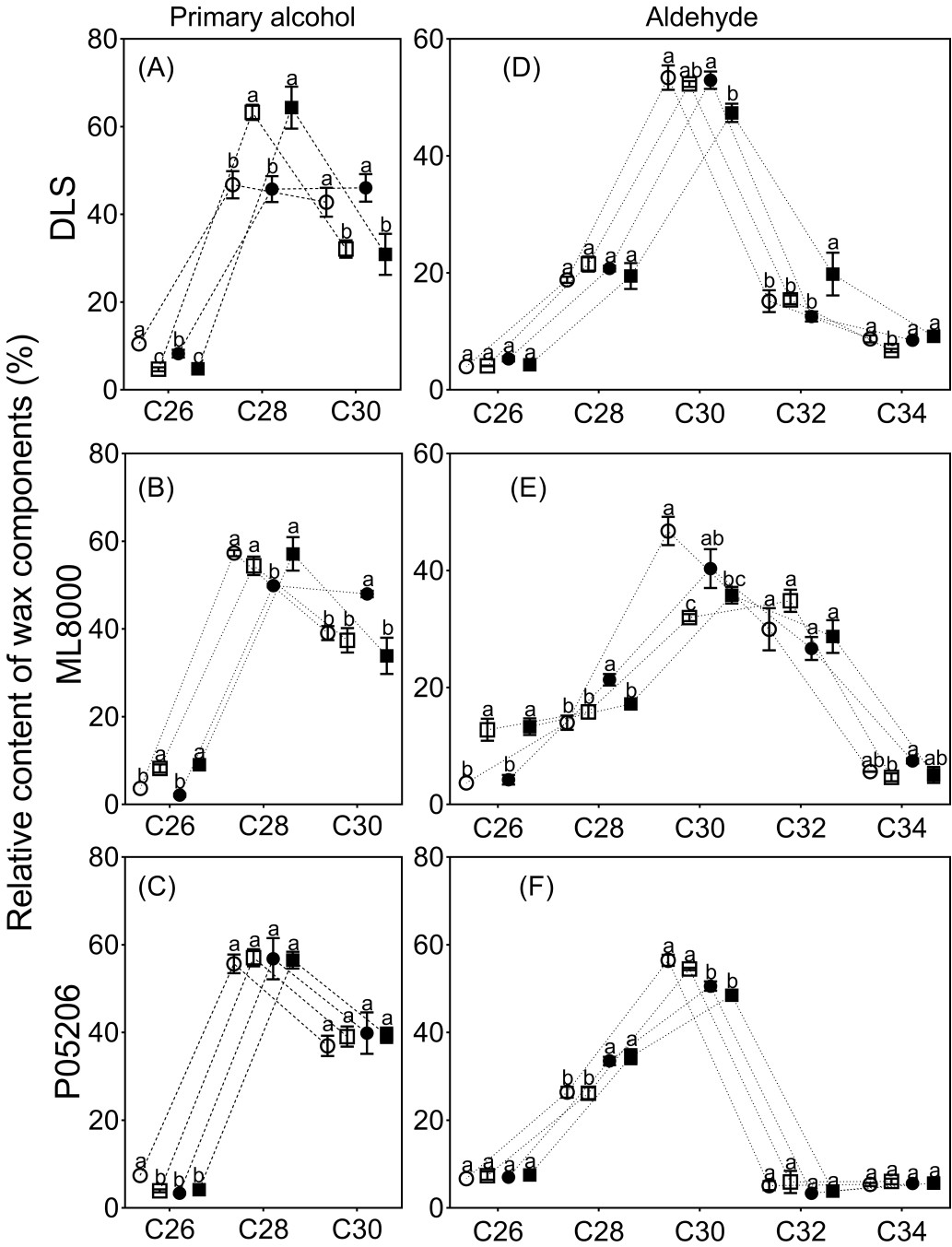

**Figure 8** **Effects of SA priming on the carbon chain distribution of wax components.** (A) (B) and (C) The chain length of n-alcohols. (D) (E) and (F) The chain length of n-aldehyde. The data were means ± SE ($n = 3$). CK, plants from non-primed seeds under well-watered condition; D, plants from non-primed seeds subjected to drought stress; SA, plants from SA-primed seeds under well-watered condition, and SA&D, plants from SA-primed seeds subjected to drought stress. Different lowercase letters within same parameters among different treatments represented significance according to the least significant difference test ($P < 0.05$). DLS, Hunnigreen; ML8000, Mule 8000; P05206, the inbred sweet sorghum.

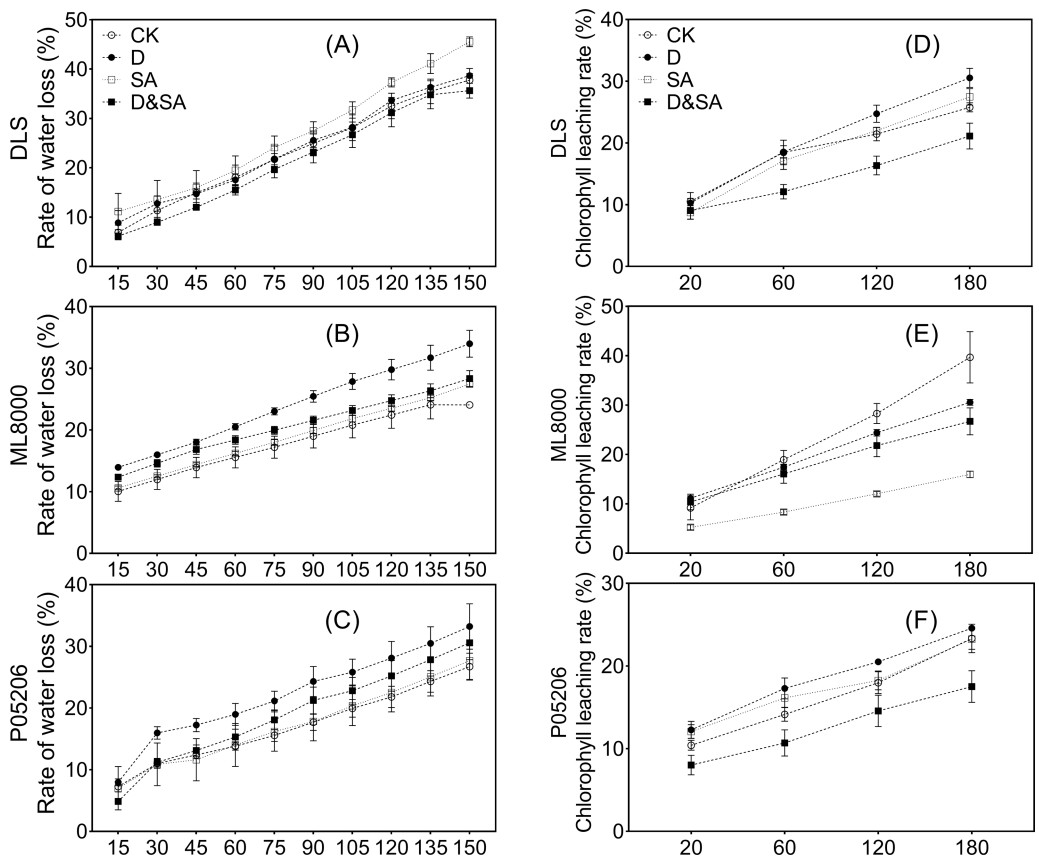

**Figure 9** **(A) (B) and (C) The leaf water loss rate (%).** (D) (E) and (F) The chlorophyll extraction rate (%). The data were means ± SE ($n = 3$). CK, plants from non-primed seeds under well-watered condition; D, plants from non-primed seeds subjected to drought stress; SA, plants from SA-primed seeds under well-watered condition, and SA&D, plants from SA-primed seeds subjected to drought stress. Different lowercase letters within same parameters among different treatments represented significance according to the least significant difference test ($P < 0.05$). DLS, Hunnigreen; ML8000, Mule 8000; P05206, the inbred sweet sorghum.

Drought stress increased the leaf water loss rate in various varieties. The results of the chlorophyll leaching rate showed that SA priming reduced the leaching rate in various varieties under both normal and drought conditions (Figs. 9D, 9E, 9F). Under drought stress, the chlorophyll leaching rate of leaves in Hunnigreen and P05206 increased.

## Correlation analysis between sorghum wax, physiological, and growth indexes

The total amount of leaf wax is significantly positively correlated with leaf water, malondialdehyde, and proline contents (Table 1). Additionally, there is also a significant positive correlation between the content of $\alpha$-amyrin and $C_{35}$-aldehyde and the biomass of sorghum. Furthermore, total wax content is significantly positively correlated with relative water content ($r = 0.79$, $P < 0.001$) in sorghum, suggesting that wax components and their carbon chain elements can also affect biomass accumulation and water content.

**Table 1  Correlation analysis between sorghum wax, physiological, and growth indexes.**

|  | Height | Biomass | Chlorophyll content | RWC | MDA | Proline |
|---|---|---|---|---|---|---|
| alkane | 0.45ns | 0.50ns | −0.51ns | 0.26ns | 0.03ns | −0.04ns |
| primary alcohol | 0.41ns | 0.54ns | −0.32ns | 0.67* | 0.29ns | 0.31ns |
| aldehyde | −0.54ns | −0.68* | 0.49ns | 0.03ns | 0.14ns | 0.49ns |
| α-amyrin | 0.68* | 0.80** | −0.48ns | 0.25ns | −0.18ns | −0.22ns |
| lanosterol | −0.05ns | 0.02ns | 0.01ns | 0.22ns | 0.41ns | 0.10ns |
| other alcohol | −0.39ns | −0.45ns | 0.12ns | 0.16ns | 0.47ns | 0.33ns |
| total wax | −0.24ns | −0.25ns | 0.20ns | 0.79** | 0.70* | 0.86** |
| C25alkane | −0.46ns | −0.38ns | 0.54ns | −0.09ns | 0.33ns | 0.15ns |
| C26alkane | −0.31ns | −0.23ns | 0.35ns | −0.01ns | 0.30ns | 0.12ns |
| C27alkane | 0.53ns | 0.58ns | −0.50ns | 0.06ns | −0.37ns | −0.19ns |
| C28alkane | −0.20ns | −0.17ns | 0.21ns | 0.17ns | 0.30ns | 0.16ns |
| C29alkane | −0.16ns | −0.29ns | 0.06ns | 0.07ns | 0.30ns | 0.06ns |
| C30alkane | 0.06ns | 0.06ns | 0.01ns | 0.28ns | 0.29ns | 0.12ns |
| C31alkane | −0.24ns | −0.13ns | 0.34ns | −0.11ns | 0.06ns | 0.03ns |
| C33alkane | −0.36ns | −0.49ns | 0.31ns | −0.24ns | −0.11ns | 0.24ns |
| C35alkane | −0.60ns | −0.66* | 0.47ns | −0.35ns | 0.10ns | 0.03ns |
| C37alkane | 0.16ns | 0.17ns | −0.02ns | −0.23ns | −0.36ns | −0.34ns |
| C26alcohol | 0.61ns | 0.58* | −0.38ns | −0.32ns | −0.85** | −0.48ns |
| C28alcohol | −0.05ns | 0.02ns | 0.27ns | 0.07ns | 0.09ns | 0.11ns |
| C30alcohol | −0.22ns | −0.28ns | −0.09ns | 0.08ns | 0.29ns | 0.11ns |
| C26aldehyde | −0.11ns | −0.04ns | 0.18ns | 0.40ns | 0.32ns | 0.38ns |
| C28aldehyde | −0.86** | −0.86** | 0.85** | −0.06ns | 0.34ns | 0.64* |
| C30aldehyde | 0.80** | 0.77** | −0.75** | −0.19ns | −0.55* | −0.79** |
| C32aldehyde | 0.49ns | 0.47ns | −0.61ns | 0.22ns | 0.07ns | −0.20ns |
| C34aldehyde | 0.09ns | 0.18ns | 0.07ns | 0.34ns | 0.15ns | 0.14ns |

**Notes.**
The asterisk in the table represents significant correlation between index.
*indicates significant correlation at 0.01 levels.
**indicates significant correlation at 0.001 levels.
ns, indicates no significant correlation; RWC, relative water content; MDA, malondialdehyde.

## DISCUSSION

### Seed priming with SA induced wax deposition improving plant drought resistance

Seed priming with salicylic acid (SA) significantly increased wax accumulation in sorghum leaves, consistent with the effects of exogenous SA on plant wax deposition observed in previous studies (*Yuan et al., 2020*). In this study, total wax content significantly increased by ∼13% under drought stress, with the effect varying by SA concentration. This variation may be attributed to the dual role of SA in modulating endogenous SA synthesis and signaling pathways (*Li et al., 2017*). Research has shown that SA priming can increase endogenous SA concentration, resulting in higher plant antioxidant capacity. This antioxidant activity is related to the expression of the key enzyme gene *ZmPAL* in SA synthesis (*Luo, Chen & Xie, 2011*). We hypothesize that a moderate endogenous SA

concentration is more beneficial for improving plant resistance, as excessive SA may disrupt cellular homeostasis. Our results support this hypothesis, as 150 mg L$^{-1}$ SA effectively accumulated proline and reduced oxidative damage to cell membranes.

### Seed priming with SA alters wax composition and activates biosynthetic pathways

Under drought stress, SA priming significantly increased the alkanes, primary alcohols, and aldehydes content in three sorghum varieties. Alkanes and aldehydes were the main wax decarboxylation pathway products, while alcohols were the main acyl reduction pathway products (*Yeats & Rose, 2013*). This suggests that SA may activate different wax synthesis genes, such as CER1, CER3, FAR, and CER4, which regulate the incorporation of long-chain fatty acids into various wax components For example, overexpression of *CER1* and *CER3* significantly increased aldehyde and alkane content in plants (*Wu et al., 2019*; *Wang et al., 2018a*; *Wang et al., 2018b*), while *FAR* and *CER* 4 overexpression increased alcohol content (*Doan et al., 2012*; *Rowland et al., 2006*). These findings align with our observation that SA priming alters wax composition, potentially through the transcriptional regulation of these genes.

### Cuticular wax enhances water retention and drought tolerance

Previous studies have shown that cuticular wax plays a role in non-stomatal transpiration under drought stress, effectively preventing water loss. In this study, the wax content in the leaves of three sorghum varieties significantly increased, and the plants thrived under drought stress. Notably, the relative water loss rate of the leaves decreased, while the relative water content increased, indicating that higher wax content can enhance drought resistance by maintaining higher water retention (*Li et al., 2020*). Furthermore, the strong correlation between wax content and proline accumulation indicates that wax deposition may enhance osmotic adjustment, a key mechanism for drought tolerance. These results demonstrate that SA-induced wax accumulation improves drought resistance by maintaining higher water retention and promoting stress-related physiological responses.

### Seed priming with SA promoted wax deposition, affecting photosynthesis in sorghum seedlings and promoting biomass accumulation

Cuticular wax deposition not only enhances drought resistance but also influences plant photosynthesis. In wheat, alterations in wax composition, such as increased alkane content or the formation of diketone wax, have been shown to improve photosynthesis and growth under drought stress (*Jiang et al., 2024*; *Su et al., 2020*). Similarly, in this study, seed priming with SA significantly increased total wax content, net photosynthetic rate, and water use efficiency. Additionally, previous studies have shown that plants sacrifice some growth to produce more protective substances to adapt to drought stress (*Wang et al., 2018a*). However, our results found that in sorghum seedlings pretreated with seed priming using SA, biomass loss was reduced, and proline content significantly increased under drought stress, indicating that SA priming mitigates the trade-off between growth and stress adaptation. This dual benefit may be attributed to the synergistic effects of wax

deposition and improved water use efficiency, which support both stress tolerance and growth.

Our findings suggest that seed priming with SA influenced the drought resistance mechanism related to cuticular wax and other growth-related resistance mechanisms. Future research should focus on elucidating the molecular mechanisms underlying SA-induced wax biosynthesis and its interaction with other stress-responsive pathways. For example, the role of key genes such as CER1, CER3, FAR, and CER4 in SA-mediated wax deposition warrants further investigation. Additionally, the potential crosstalk between SA signaling and other phytohormones, such as abscisic acid (ABA) and jasmonic acid (JA), could provide insights into the broader regulatory network governing drought tolerance.

## CONCLUSIONS

We investigate the potential of SA-priming to enhance drought tolerance in sorghum seedlings. When sorghum plants derived from SA-primed seeds encounter drought stress, proline and wax accumulation work synergistically to regulate growth and enhance drought adaptation, ultimately contributing to drought tolerance without penalizing growth. Identifying 150 mg L$^{-1}$ SA as optimal for drought resilience offers a practical, cost-effective solution to enhance crop productivity and sustainability in water-limited regions, with significant implications for global food security and resource conservation. Future research should focus on optimizing priming techniques and exploring their underlying mechanisms to further improve crop resilience under water-limited conditions.

## ACKNOWLEDGEMENTS

We are deeply grateful to Mr. Sennan Li for his improvement of the Materials & Methods section. Meanwhile, we would like to express their gratitude to EditSprings for the expert linguistic services provided.

### Funding

This work was supported by the Scientific Research Foundation of Hainan Tropical Ocean University (No. RHDRC202318). The funders had no role in study design, data collection and analysis, decision to publish, or preparation of the manuscript.

### Grant Disclosures

The following grant information was disclosed by the authors:
Scientific Research Foundation of Hainan Tropical Ocean University: RHDRC202318.

### Competing Interests

The authors declare there are no competing interests.

## Author Contributions

- Luhua Yao conceived and designed the experiments, performed the experiments, analyzed the data, prepared figures and/or tables, authored or reviewed drafts of the article, and approved the final draft.
- Yitao Wu performed the experiments, prepared figures and/or tables, and approved the final draft.

## Data Availability

The raw measurements are available in the Supplementary File.

## Supplemental Information

Supplemental information for this article can be found online at http://dx.doi.org/10.7717/peerj.20014#supplemental-information.

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
