# Peer review of "Effects of seed priming with salicylic acid on cuticular wax deposition in sweet sorghum under drought stress"

_PeerJ, doi:10.7717/peerj.20014_

## Round 0.1 · original submission · Major Revisions

Authors should work on 'statistical analysis', clear presentation of 'methodology', 'experimental design', 'data analysis' and 'figures' to improve the manuscript.

·

Basic reporting

No comment.

Experimental design

Line 144: "One-way ANOVA was used:” I wonder why one way while you studied three factors, and the figures shows the interaction between them. It’s really confused, please fix or explain that through text body too.

Validity of the findings

Data analysis looks confused, author should explain that. see other comments.

Additional comments

1. Don't mention abbreviations in the title, please.
2. Don't repeat words from the title in keywords, please.
3. Line 13 to 17: “Three sweet sorghum varieties were treated with different water levels (normal and drought) and SA concentrations (50, 100, 150, 200, 250 mg L).” you need to add the experimental design and replicates, please.
4. Line 15: (normal and drought), The expression sounds unclear! Change it to levels or else.
5. Line 71 to 74: Delete or move this paragraph to the material and methods. Please. “In this study, we tested three sweet sorghum varieties, applying five SA concentrations (50, 100, 150, 200, and 250 mg L-1) under two moisture conditions: 80% normal moisture and 35% Drought treatment”.
6. Line 88 to 89: you need to tell very precise details about treatments “seedlings were either exposed to drought or treated with exogenous hormones”
7. Line 319 to 325 refer to results! It should not be in conclusion. Please, rephrase the conclusion and delete any result from it. The focus can be on physiological changes in the traits studied rather than changes in their numerical value.
8. Figures 2 to 8 display resolution needs to be improved.

Reviewer 2 ·

Basic reporting

The manuscript presents a well-designed study with valuable insights into SA priming for drought tolerance. Addressing statistical rigor, clarifying methodological details, and refining the discussion of mechanisms would enhance its impact. The findings are novel and relevant to crop science, meriting publication after minor revisions.

Experimental design

Clarify Treatment Groups: The description of treatment groups (CK, D, SA, SA&D) does not explicitly include water-primed controls, which were mentioned in the Methods. Ensure consistency between the experimental design and the analyzed groups.

Statistical Analysis: A two-way ANOVA would better account for interactions between SA concentration and water stress, rather than the one-way ANOVA used. This would strengthen the interpretation of synergistic/antagonistic effects.

Replicates and Sample Size: While n=4 is acceptable, increasing replicates (e.g., n=6) could enhance statistical power, especially for variability across genotypes.

Validity of the findings

Concentration Justification: The rationale for selecting SA concentrations (50–250 mg/L) should be explicitly linked to prior studies (e.g., equivalence to µM/mM ranges used in other crops).

Statistical Reporting:

Address discrepancies in Table 1: The note mentions "" for p<0.05 and "" for p<0.01, but some entries have "," which is undefined. Standardize significance notation.

Clarify whether correlations were adjusted for multiple comparisons (e.g., Bonferroni correction) to reduce false positives.

Mechanistic Gaps: While the discussion references genes like CER1 and CER3, the study does not measure gene expression or enzyme activity. Suggest future work to validate these mechanisms.

Additional comments

Clarity and Language:

Repetition: The abstract and conclusion are redundant. Condense the conclusion to highlight key findings without repeating the abstract.

Definitions: Define terms like "resistance index" and clarify how metrics like "water use efficiency" were calculated.

Methodological Details:

Drought Stress Protocol: Specify the duration and method of drought application (e.g., soil moisture reduction to 35% field capacity over X days).

Priming Protocol: Clarify whether water priming was included as a control in all analyses (e.g., in Figures 1–8).

Figures and Tables:

Figure Legends: Ensure all axes and abbreviations (e.g., DLS, ML8000) are clearly defined.

Data Availability: Confirm raw data for all figures/tables are included in supplemental files, as required by PeerJ.

Broader Impact:

Highlight the practical implications of identifying 150 mg/L SA as optimal for drought resilience, particularly for agricultural applications in water-limited regions.

·

Basic reporting

The manuscript has many typos. Need to seriously edit. The English writing also needs to improve.

In the introduction, authors need to present the effects of SA supplementation in drought conditions. And authors need to clarify the novelty of the manuscript. Besides, some recent SA effects on abiotic stress need to be discussed.

The resolution of the figure needs to improve.

Experimental design

In abstract, Its huge. Need to be short and concise. The authors need to rewrite the result section in abstract precisely. Also need to add some numerical values.

Please don’t start the sentences with abbreviations. For example, line 98. Check throughout the manuscript.

Why were these commercial varieties used? Needs to clarify.

Validity of the findings

The orientation of the result section is very poor. Authors need to rewrite the “Seed priming improved growth and drought tolerance of sorghum seedlings” with small headings so that readers can easily understand. In this section, they wrote too much. Photosynthetic components, RWC, and proline can be separated.

Discussion needs to improve; why do authors discuss only under two headings? They need to compare their results with existing literature and make some assumptions about why increment or decrease.

Conclusion needs future recommendations.

---

## Round 0.2 · Minor Revisions

Manuscript format: in Introduction and M+M section, you use double space between the lines and single in Results and Discussion. Please correct the second part by using double-spaced lines.

Abstract
L27: “stress memory” appears only here and it is not determined in any way or discussed in any section of the MS. Please remove it from the Abstract or provide relevant measurements/determination and explanations in all the sections of the MS.

Your Conclusions section (L350-358) is well written and justified from your results, so you can briefly refer to them also in the Conclusions part of your Abstract (after removing the stress memory sentence).

I think it is better to mention in the Abstract that 3 varieties were tested. This will explain to the reader the range of the numbers you give in the Results part of the Abstract.

Introduction
Please be consistent in the format: in L71-87 make the references and the scientific name of the plants in italics (L71).

L84. Delete the “

Materials and Methods
This section needs more detailed description of the Methods used for all the determinations.

My general comment: in each instrument used, add the manufacturer (e.g LC-pro SD, (ADC Bioscientific, London, UK). This should be added also in GC and GC/MS and photosynthesis.

My specific comments follow:
1) Leaf wax extraction: which were your standards and the procedure in order to identify the substances in GC and GC/MS. Add relevant information.

2) % of water loss: in L153-154 please use the same abbreviations as in RWC formula.

3) Chlorophyll extraction rate: You do not describe in M+M section any method of determining such a parameter. Additionally, in Abstract you use the term “Chlorophyll leaching rate” and in the Results and Figure 8 you use the term “Chlorophyll extraction rate”. I guess that you may want to refer to the first one which is more valid, so please:
a) Add the relevant part in the M+M with all the details under the sub-title: Determination of chlorophyll leaching rate
b) replace all the “Chlorophyll extraction rate” with “Chlorophyll leaching rate” throughout the MS. Please be sure that you also corrected the caption and Y-axis name of the relevant figure.

4) Proline and malondialdehyde contents: describe in more details -but briefly- the procedure and not only a simple reference to the main solvent and the relevant Literature.

Discussion
L295-296: this 30% increase is valid for the last two varieties and not for DLS. Please add this distinction.

Figures
Figure 1: X-axis title: SA concentration (μg L-1)
Figure 2: correct the X-axis title as in Figure 1.
Figure 3: use the abbreviations (e.g. Gs), and not both the whole term AND the abbreviation.

Figure 4: Correct the figure caption as follows: “Effects of SA priming with different concentrations on chlorophyll content (SPAD value), relative water content and water use efficiency”.
Additionally, make the presentation of the figure at landscape orientation, to be clearer and easier to read.

Figure 5: correct the X-axis title as per Figure 1.
Also, delete all the “The” from all Y-axis titles. After that, capitalize the first letter, e.g. “Proline content (μg g FW-1).

Figure 6: Capitalize the first letter of Y-axis title.

Figure 7: Delete the “The” from Y-axis title and again present the figure in landscape orientation to be bigger and clearer.

Figure 8. Delete the gaps between letters in the figure caption. Delete the “The” from Y-axis title.
Figure 8: are you sure about these very low values of water loss, given that they are %?

·

Basic reporting

Dear authors,
Please review the manuscript grammatically and linguistically, pay attention to the accuracy of the figures professionally and their presentation proficiently (trying landscape, not portrait, show, maybe), and review the extent to which you have complied with all reviewers' comments without exception. We do appreciate that you have come a long way in improving the manuscript, and you were truly committed to this, and there is primarily very little left. You still need to work on the notes above.
Best regards

Experimental design

No comment.

Validity of the findings

No comment.

Additional comments

No comment.

Reviewer 2 ·

Basic reporting

no comment

Experimental design

no comment

Validity of the findings

no comment

Additional comments

accept as it is

·

Basic reporting

Thank you, authors, for revising the manuscript according to the comments. The manuscript is much improved. It can be accepted.

Experimental design

Thank you, authors, for revising the manuscript according to the comments.

Validity of the findings

Thank you, authors, for revising the manuscript according to the comments.

---

## Round 0.3 · Minor Revisions

You have carefully addressed most of my comments. I found good idea the splitting of Figure 7!

Please correct the following:
1) Include the "SA" in the X-axis of Figs 2, 3, 5.
So the X-axis title will be "SA concentration (μ L-1)".

2) Do the same for all Y-axis of Figure 4.

3) Abstract: I think that the way the following text is written "three sweet sorghum seeds were primed" may confuse the reader. It may correspond to 3 (count) seeds and not to 3 varieties. So please correct it to "sweet sorghum seeds belonging to three varieties were primed".

---

## Round 0.4 · Minor Revisions

We are resending the decision from May 26, 2025 (2.0) to resolve your inability to resubmit.

You have carefully addressed most of my comments. I found good idea the splitting of Figure 7!

Please correct the following:
1) Include the "SA" in the X-axis of Figs 2, 3, 5.
So the X-axis title will be "SA concentration (μ L-1)".

2) Do the same for all Y-axis of Figure 4.

3) Abstract: I think that the way the following text is written "three sweet sorghum seeds were primed" may confuse the reader. It may correspond to 3 (count) seeds and not to 3 varieties. So please correct it to "sweet sorghum seeds belonging to three varieties were primed".

---

## Round 0.5 · accepted · Accept

All issues addressed successfully.